# Predicting yield performance of parents in plant breeding: A neural collaborative filtering approach

**Saeed Khaki**[1]*, **Zahra Khalilzadeh**[2], **Lizhi Wang**[1]

**1** Industrial and Manufacturing Systems Engineering Department, Iowa State University, Ames, Iowa, United States of America, **2** Institute for Transportation, Iowa State University, Ames, Iowa, United States of America

* skhaki@iastate.edu

## Abstract

Experimental corn hybrids are created in plant breeding programs by crossing two parents, so-called inbred and tester, together. Identification of best parent combinations for crossing is challenging since the total number of possible cross combinations of parents is large and it is impractical to test all possible cross combinations due to limited resources of time and budget. In the 2020 Syngenta Crop Challenge, Syngenta released several large datasets that recorded the historical yield performances of around 4% of total cross combinations of 593 inbreds with 496 testers which were planted in 280 locations between 2016 and 2018 and asked participants to predict the yield performance of cross combinations of inbreds and testers that have not been planted based on the historical yield data collected from crossing other inbreds and testers. In this paper, we present a collaborative filtering method which is an ensemble of matrix factorization method and a neural network to solve this problem. Our computational results suggested that the proposed model significantly outperformed other models such as deep factorization machines (DeepFM), generalized matrix factorization (GMF), LASSO, random forest (RF), and neural networks. Presented method and results were produced within the 2020 Syngenta Crop Challenge.

**Data Availability Statement:** The data analyzed in this study was provided by Syngenta AG company for 2020 Syngenta Crop Challenge. We accessed the data through annual Syngenta Crop Challenge. During the challenge, September 2019 to January

## Introduction

Plant breeding is an important scientific area which helps increase the food production. One of the important decisions faced by plant breeders is the selection of the appropriate parents for artificial crosses [1]. The degree of success achieved by plant breeding programs highly depends on the selection of the best parent combinations and it is crucial that genetic variability be present in the progenies since populations with reduced genetic potential may result in a waste of time and money [1]. Historically, plant breeders test new experimental hybrids in a diverse set of locations and measure their performance, then select the highest yielding hybrids. The process of selecting the correct parent combinations and testing the experimental hybrids is highly time-consuming, simply due to the large number of potential parent combinations to create and test [2].

2020, the data was open to the public. Data cannot be shared publicly because of non-disclosure agreement. Data are available from the Syngenta (contact via https://www.ideaconnection.com/syngenta-crop-challenge/challenge.php) for researchers who meet the criteria for access to confidential data.

**Funding:** This work was partially supported by the National Science Foundation under the LEAP HI and GOALI programs (grant number 1830478) and under the EAGER program (grant number 1842097). The funder had no role in study design, data collection and analysis, decision to publish, or preparation of the manuscript.

**Competing interests:** The authors have declared that no competing interests exist.

Many statistical approaches have been used to help breeder select appropriate parents for crosses. Barbosa-Neto et al. used genetic relationship as a predictor of the relative performance of hybrid combinations which led to reduced time and cost of hybrid testing [3]. Van Beuningen and Busch analyzed genetic diversity among wheat cultivars using cluster analysis to quantify diversity which is important for plant breeders [4]. Mixed models were also employed at different stages of plant breeding programs such as crossing-progeny tests and multi-environment yield trials, where they are used to predict cross performance of untested crosses and study genotype-environment interactions in crossing-progeny tests and multi-environment yield trials stages, respectively [1, 5]. Bernardo et al. used best linear unbiased prediction (BLUP) method for predicting the performance of untested crosses in corn hybrids [6]. Panter and Allen employed the BLUP method for identifying superior soybean cross combinations [7]. Balzarini applied a mixed model-based approach for sugarcane cross prediction [8]. Regression methods such as Ridge [9] is also used for the cross prediction performance. For example, Hofheinz el al. proposed genome-based prediction of test cross performance using Ridge regression [10].

More recently, machine learning techniques have been applied to genomic selection (GS) and plant breeding, including support vector machines (SVM), random forest, and artificial neural networks (ANN). Machine learning models consider the output as an implicit function of the input variables. Montesinos-López et al. evaluated prediction performance of SVM, ANN, and BLUP models in the GS context and found that SVM and ANN models had a comparable performance with the BLUP model [11]. González-Camacho et al. applied several machine learning methods such as random forest, neural networks, and SVM to predict rust resistance in wheat and found that SVM had the best overall genomic based prediction performance [12]. González-Camacho et al. used probabilistic neural network for genome-based prediction of corn and wheat [13]. Machine learning approaches were also used for predicting performance of crops under different environmental conditions [14–19].

In this paper, we use a neural network based collaborative filtering method to predict the yield performance of cross combinations of inbreds and testers that have not been planted based on the historical yield data collected from crossing other inbreds and testers together. The proposed model is a hybrid one which combines a neural network and matrix factorization [20] method. Our proposed model takes in the inbred ID, tester ID, location ID, and genetic grouping ID as the inputs and collaboratively predicts the yield of cross combinations through learning the behavior of all inbreds and testers. The proposed model is able to accurately predict the average yield performance of untested cross combinations which would be useful in the selection of best artificial crosses. To the best of our knowledge this is the first study to use the neural network based collaborative filtering method in plant breeding for cross yield prediction.

The remainder of this paper is organized as follows. Problem statement section defines the research problem. Methodology section provides a detailed description of the proposed method. Design of experiments section describes the computational experiments. Results section explains the results. Analysis section provides the analysis performed based on the proposed model. Finally, we conclude the paper in discussion section.

## Problem statement

New experimental hybrids are created in plant breeding programs by making crosses of two crop parents. Whether these crosses will produce better hybrids (e.g., with higher yields and better adaptability to regional soil and weather conditions) depends on the quality of the crosses. It would be prohibitive to enumerate all the biparental combinations, which need to be carried out and tested in multiple environments through a time-consuming and labor

intensive process [2]. In practice, plant breeders can only afford to make a small subset of crosses, create new experimental hybrids, plant them in different environments to assess their performances, and then select the best hybrids therein.

In the 2020 Syngenta Crop Challenge [2], participants were challenged to use real-world data to predict the performance of potential biparental crosses, which would be beneficial for plant breeders to identify desirable crosses and new hybrids. The underlying research problem is to predict the yield performance of all possible inbred-tester combinations based on historical yield data of those crosses that have actually been made.

The total number of biparental combinations is 294,128, which is 593 inbreds by 496 testers. Historically observed yields of 3.71% of these corn hybrids were provided resulted from 10,919 unique cross combinations, where included 199,476 observations of hybrids planted across 280 different locations between 2016 and 2018. The dataset also included 14 genetic groups of all inbreds and testers, representing genetic similarity of the parents.

## Methodology

We designed a neural collaborative filtering method to predict the yield of inbred-tester combinations based on the historical data of hybrids actually produced and planted. This model takes the inbred ID, tester ID, location ID, and genetic grouping ID as the inputs, and it learns the inbred-tester interactions through collaborative filtering methods while treating the location and genetic grouping data as auxiliary information. Fig 1 shows the structure of the proposed model.

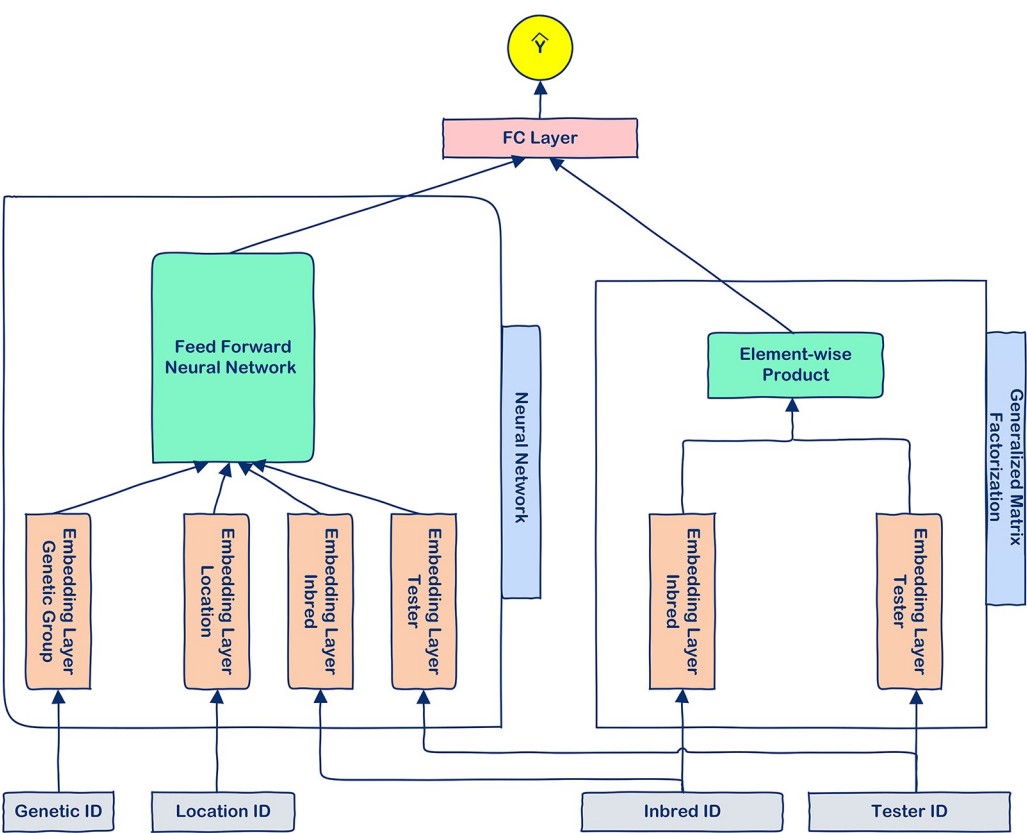

**Fig 1. The modeling structure of the proposed model.** The proposed model consists of a neural network component and a generalized matrix factorization component, which are integrated by a fully-connected (FC) layer.

The proposed method is a hybrid one that combines two complementary components. The neural network component was designed to learn high-order interactions between inbreds and testers, whereas the generalized matrix factorization (GMF) component was designed to learn low-order feature interactions as in [21]. Embedding layers were used to improve the effectiveness of the two components by converting sparse one-hot inputs to dense vectors. To achieve the highest learning capability of the model, we trained the "embedding layer tester" and "embedding layer inbred" separately for the two components. Our method is inspired by the model proposed by He et al. [21], but has the following main differences: (1) our method is for explicit feedback recommendation while the model proposed by He et al. is for implicit feedback recommendation, (2) our method includes auxiliary information, but the model proposed by He et al. does not use any auxiliary information and just focuses on the interactions between users and items, (3) the loss functions of proposed methods are also different.

**Input and embedding layers**. Given an inbred-tester pair $(b, t)$ and their corresponding genetic grouping $(g)$ and planting location $(l)$, we perform one-hot encoding on their IDs to get one-hot feature representation vectors, denoted as $x_b \in \mathbb{B}^{n_b \times 1}$, $x_t \in \mathbb{B}^{n_t \times 1}$, $x_g \in \mathbb{B}^{n_g \times 1}$, $x_l \in \mathbb{B}^{n_l \times 1}$, respectively, where $n_b$, $n_t$, $n_g$, and $n_l$ denote the total number of inbreds, testers, genetic groups, and locations, respectively. Let $W_1^b \in \mathbb{R}^{n_b \times k_1}$, $W_2^b \in \mathbb{R}^{n_b \times k_2}$, $W_1^t \in \mathbb{R}^{n_t \times k_1}$, $W_2^t \in \mathbb{R}^{n_t \times k_2}$, $W^g \in \mathbb{R}^{n_g \times k_g}$, and $W^l \in \mathbb{R}^{n_l \times k_l}$ be the embedding matrices for inbreds in GMF component, inbreds in the neural network component, tester in the GMF component, tester in the neural network component, genetic groupings, and locations, respectively, where $k_1$, $k_2$, $k_g$, and $k_l$ denote the number of latent factors used in each matrix. These embedding layers map sparse one-hot inputs to dense vectors, which can be seen as the latent vectors. For an inbred-tester pair $(b, t)$ with genetic grouping $(g)$ planted in location $(l)$, the latent (dense) vectors are obtained as below:

$$
\begin{aligned}
d_1^b = x_b^T \cdot W_1^b, \quad d_2^b = x_b^T \cdot W_2^b, \quad d_1^t = x_t^T \cdot W_1^t \\
d_2^t = x_t^T \cdot W_2^t, \quad d^g = x_g^T \cdot W^g, \quad d^l = x_l^T \cdot W^l
\end{aligned}
\tag{1}
$$

**Generalized matrix factorization**. The generalized matrix factorization (GMF) is a generalization of matrix factorization method [22] which aims to learn low-order interaction between inbreds and testers. Let inbred latent vector and tester latent vector for an inbred-tester pair $(b, t)$ be $d_1^b$ and $d_1^t$, respectively. Then, the following mapping is defined to capture low-order feature interactions:

$$
f_1(d_1^b, d_1^t) = d_1^b \odot d_1^t
\tag{2}
$$

Here $\odot$ denotes element-wise product of vectors. Such mapping can be used directly for predicting the yield of the cross combination of the inbred-tester pair $(b, t)$ which is as follows:

$$
\hat{y}_{bt} = (d_1^b \odot d_1^t) \cdot h
\tag{3}
$$

Here $h \in \mathbb{R}^{k_1 \times 1}$ is the weights of the output layer. If the h is enforced to be 1, then the model is exactly matrix factorization model. Since GMF does not enforce the weights to be 1 and learns these weights form data, it lets varying importance of latent dimensions which would improve the accuracy of results [21].

**Neural network**. The neural network component of the model aims to learn the high-order interaction between the inbreds and testers. Let $d_2^b$, $d_2^t$, $d^g$, and $d^l$ be the inbred latent vector, tester latent vector, genetic grouping latent vector, and location latent vector, respectively.

Given the network has $J$ layers, the neural network model is defined as follows:

$$a_1 = \phi_1(W_1[d_2^b \ d_2^t \ d^g \ d^l]^T + b_1)$$
$$a_2 = \phi_2(W_2 \ a_1 + b_2)$$
$$\ldots\ldots$$
$$a_J = \phi_2(W_J \ a_{J-1} + b_J)$$

(4)

Here $W_j$, $b_j$, $\phi_j$, and $a_j$ denote the weight matrix, bias vector, activation function, and the output of $j$th layer of the network, respectively. We used embedded dense vectors of data as the input of the neural network model rather than the sparse one-hot vectors to make the network easier to train [23].

**Combining the GMF and neural network**. Although each component, namely GMF and the neural network could be used individually for the prediction, we combine the two components to capture both low-order and high-order interactions between inbreds and testers. Let $q_{GMF}$ and $q_{NN}$ denote the output of the generalized matrix factorization and the neural network components, respectively. Then, the prediction of the model for an inbred-tester pair $(b, t)$ is defined as below:

$$\hat{y}_{bt} = W^T \begin{bmatrix} q_{GMF} \\ q_{NN} \end{bmatrix} + b$$

(5)

Here the $W$ and $b$ are the weight matrix and bias vector of the final output layer, respectively.

## Design of experiments

We used the following hyperparameters to train the proposed model. All embedding layers have 32 latent factors. We tried different number of latent factors in the embedding layers such as 8, 16, 32, and 50 and found that 32 latent factors resulted in the best overall performance. We found that using the same number of latent factors in all embedding layers improved the overall performnace of the model as recommended in [23]. The neural network part of the model has 3 layers with a tower network architecture. The first, second, and third layers of the network have, respectively 64, 32, and 16 neurons. We tried different types of activation functions, including ReLU, sigmoid, and tanh and found that ReLU resulted in the most accurate predictions. To avoid overfitting, we used dropout [24] with the keep probability of 0.7 after each fully-connected layer of the network. All weights of the network were initialized with Xavier method [25]. Stochastic gradient descent (SGD) were used with mini-batch size of 16. We used Adam optimizer [26] with learning rate of 0.03% to minimize the loss function, which was Huber loss in our study. As shown in Eq 6, Huber loss combines both squared loss and absolute loss. The value of $\delta$ in our study is set to be 0.1. The model was trained for the maximum of 70,000 iterations. The proposed model was implemented in Python using the Tensorflow library [27].

$$L_\delta(y, \hat{y}) = \begin{cases} 0.5 \ (y - \hat{y})^2 & \text{if } |y - \hat{y}| \leq \delta \\ \delta \ |y - \hat{y}| - 0.5 \ \delta^2 & \text{otherwise} \end{cases}$$

(6)

Here $y$ and $\hat{y}$ are the observed and the predicted values, respectively.

**Pre-training**. Since the loss function of the model is non-convex and high-dimensional, it is hard to optimize and we can only find a local-optimal solution for it in practice through gradient-based optimization methods [25]. The performance of these gradient-based optimization methods considerably depends on the quality of the initialized values for deep learning models [28]. As such, to further improve the prediction accuracy of the proposed model, we initialized the proposed model with the pre-trained embedding layers. We first trained the GMF and the neural network models separately with Xavier initializations. Then, we used their trained embedding layer parameters as the initialization for the corresponding parts in the proposed model.

To evaluate the performance of the proposed model, we compared the proposed model with the following models:

- **DeepFM**: this model proposed by [23] combines factorization machines (FM) and a neural network to capture both low-order and high-order feature interactions with no need of manual feature engineering. The following hyperparameters were used in the DeepFM model which resulted in the best overall performance. All embedding layers have 32 latent factors. Dropout with the keep probability of 0.6 was used to prevent overfitting. The neural network part of the model has 2 layers, each having 32 neurons. ReLU activation functions were used in the neural network part of the model as recommended in the original model. All weights were initialized with Xavier method and the model was trained using SGD.

- **Random Forest**: random forest [29] is an ensemble learning method which is robust against overfitting. Random forest is considered as a non-parametric model and learns the underlying function completely from data. Different number of trees were tried and we found that 150 trees resulted in the best overall performance. We also found that maximum depths of 15 for the trees led to the most accurate results.

- **Neural Network**: to examine the performance of the neural network part of the model, we used this part of the model separately for prediction.

- **GMF**: to examine the performance of the generalized matrix factorization part of the model, we used this part of the model separately for prediction.

- **FM**: to examine the performance of the factorization machines part of the DeepFM model, we used this part of this model separately for prediction. FM model originally proposed by [20] captures pairwise feature interactions.

- **Proposed Model without Pre-training**: to evaluate the effect of pre-training on the performance of the proposed model, we trained model without pre-training from random initialization.

- **LASSO**: LASSO [30] is used as a benchmark model for the comparison between linear and nonlinear models. We tried different values for the $L_1$ coefficients in the LASSO model and found that 0.8 resulted in the most accurate predictions.

## Results

To completely evaluate the performance of the competing models, we used two testing procedures which are described below:

- **Cross validation**: we evaluated the performance of the proposed model using 10-fold cross validation which resulted in having 19,947 observations in each fold.

**Table 1. Yield prediction performance of the competing models using 10-fold cross validation.**

| Model | Training RMSE | Training Correlation Coefficient (%) | Test RMSE | Test Correlation Coefficient (%) |
|---|---|---|---|---|
| Proposed Model with Pre-training | 0.0942 | 44.21 | 0.0962 | 39.76 |
| Proposed Model without Pre-training | 0.0913 | 49.86 | 0.0979 | 36.08 |
| DeepFM | 0.0878 | 54.53 | 0.0998 | 35.82 |
| Random Forest | 0.1016 | 25.01 | 0.1025 | 20.71 |
| Neural Network | 0.0947 | 44.05 | 0.0986 | 33.84 |
| GMF | 0.0953 | 41.83 | 0.0995 | 32.21 |
| FM | 0.0926 | 46.92 | 0.1003 | 33.22 |
| LASSO | 0.1047 | 0.0 | 0.1047 | 0.0 |

The mean±standard deviation of the yield is 1.002±0.1047. The GMF and FM stand for the generalized matrix factorization and factorization machines, respectively.

- **Hold-out test**: we randomly selected 1,042 cross combinations of inbreds and testers and used it as test data. The training data included all observations in the dataset excluding observations corresponding to selected cross combinations. Since the planting locations were not provided in the test data in the 2020 Syngenta Crop Challenge, we did the following analysis. First, we predicted the yield of each cross combination of inbreds and testers across all 280 locations. Then, we took the average of all predictions as a final prediction. We defined the response variables for the hold-out test data as the average yield of cross combinations across planting locations.

Tables 1 and 2 compares the 10-fold cross validation and hold-out performances of the models on both training and test datasets with respect to the root-mean-squared-error (RMSE) and correlation coefficient, respectively.

The results suggest the effectiveness of the proposed model in predicting the yield performance of cross combinations of inbreds and testers based on the historical hybrid data collected from crossing of other inbreds and testers together. The proposed model significantly outperformed other models to varying extent. DeepFM had a better performance compared to other competing models except for the proposed model with respect to correlation coefficient based on the 10-fold cross validation results. DeepFM outperformed LASSO, FM and random forest with respect to all performance measures due to capturing both low-order and high-order feature interactions. However, DeepFM did not outperform the neural network and

**Table 2. Yield prediction performance of the competing models using hold-out test.**

| Model | Hold-out Test RMSE | Hold-out Test Correlation Coefficient (%) |
|---|---|---|
| Proposed Model with Pre-training | 0.0256 | 90.46 |
| DeepFM | 0.0515 | 44.52 |
| Random Forest | 0.0550 | 23.65 |
| Neural Network | 0.0481 | 53.07 |
| GMF | 0.0284 | 87.42 |
| FM | 0.0580 | 40.40 |
| LASSO | 0.0567 | 0.0 |

The GMF and FM stand for the generalized matrix factorization and factorization machines, respectively.

GMF with respect to RMSE. The performance of the LASSO was weak due to its linear modeling structure which could not capture the nonlinear interactions between inbreds and testers. Random forest performed better than LASSO due to its nonlinear modeling structure which was able to learn the nonlinear interactions between inbreds and testers. The overall performances of LASSO and random forest were poor compared to other models since they are not well-suited for highly sparse data.

The GMF had a comparable performance with the neural network while both completely outperformed the LASSO and random forest due to capturing interactions between inbreds and testers which is of great importance in the neural collaborative filtering. FM had a similar performance to the neural network and GMF with respect to correlation coefficient based on the 10-fold cross validation results, but its performance was not comparable to the neural network and GMF based on the hold-out test results. FM performed better than LASSO and random forest due to capturing first and second order interactions between testers and inbreds. One of the main reasons of the higher prediction accuracy of the GMF, FM, and the neural network compared to the LASSO and random forest is the use of embedding layers which map sparse one-hot inputs to dense vectors, which can be seen as the latent vectors. Since the proposed model is the ensemble of the GMF and the neural network, it can learn both low-order and high-order interactions which resulted in the highest prediction accuracy compared to other models.

The proposed model performed better than DeepFM model because it captures the interactions between inbreds and testers and considers the location ID and genetic grouping ID as auxiliary information. As a result, the proposed model is able to focus more on the interactions between testers and inbreds. Moreover, DeepFM model uses shared embedding layers between FM and the neural network parts, whereas the proposed model which uses separate embedding layers between GMF and the neural network parts. As such, shared embedding would limit the capacity of the DeepFM model to learn the feature interactions. The performance comparison between the proposed model with pre-training and the proposed model without pre-training suggests that pre-training improved the prediction accuracy of the proposed model. The prediction accuracies on the hold-out test data are higher compared to the cross validation accuracies because the response variable for the hold-out test data is the average yield of the cross combinations.

## Analysis

**Yield prediction performance without fine details**. Hammer et al. [31] suggested using coarse-grained models for phenotypic prediction in plant breeding without including much of the fine detail information. Inspired by their suggestion, we excluded the fine details such as inbred ID and tester ID from the proposed model and used just genetic grouping ID and location ID for the yield prediction which are considered high-level information. Table 3 shows the yield prediction performance of the proposed model without including inbred ID and

**Table 3. Yield prediction performance of the proposed model without including inbred ID and tester ID in the model.**

| Performance Evaluation Method | Test RMSE | Test Correlation Coefficient (%) |
|---|---|---|
| 10-fold Cross Validation | 0.1015 | 25.26 |
| Hold-out Test | 0.0263 | 89.85 |

The RMSE and correlation coefficient shows the performance on the test data.

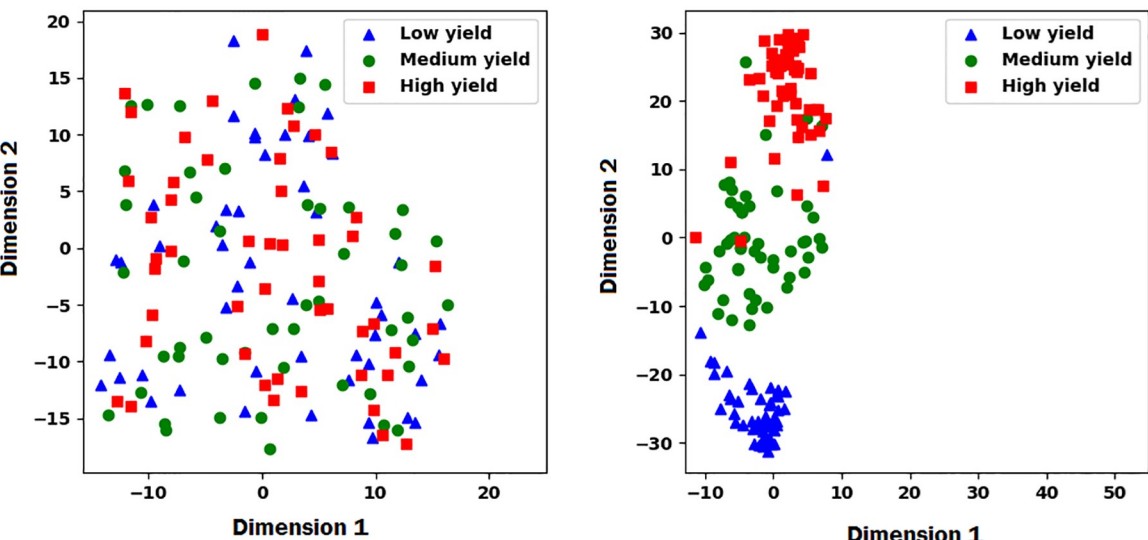

**Fig 2. The left and right plots show the t-SNE embedded plot of inbred latent factors before and after training, respectively.** The plots illustrate how the proposed model was able to differentiate the high, medium, and low yield inbreds based on their corresponding latent factors in the embedding layer.

tester ID in the model. As shown in Table 3, the yield prediction performance of the proposed model without using inbred ID and tester ID did not drop significantly compared to corresponding results in Tables 1 and 2 which suggested most of the variation in the crop yield is explained by the genetic grouping and the environmental factors. The results also indicated that a high level model based on only genetic grouping ID and location ID might be potentially useful for the yield prediction.

**Visualization of embedding layers**. To better understand how the proposed model is collaboratively learning to predict the performance of the cross combinations of inbred and testers, we used t-distributed stochastic neighbor embedding (t-SNE) [32] to visualize the embedding layers for inbreds and testers of the proposed model. First, we found the marginal yield of each inbred by averaging across all corresponding observed yield of cross combinations. Then, we categorized all inbreds into high, medium, and low yield based on their marginal average yield. We selected 50 individuals from each category and used t-SNE method to transfer the high dimensional latent factors of embedding layers to the 2-dimensional space for visualization. We performed the same process for the testers. Results are shown in Figs 2 and 3, which reveal that the proposed model was able to to differentiate high, medium, and low yield inbreds and testers using their corresponding latent factors.

**Decision making based on the proposed model**. The goal of this study was to find the best possible cross combinations of inbreds and testers for crossing. Thus, we demonstrate how our proposed model can be used to make informed crossing decisions using the following analysis. We predicted the yield of each cross combination of inbreds and testers across all 280 locations. Then, we took the average of all predictions across all locations and considered it as the average yield performance of the corresponding inbred and tester combination. We performed the above analysis for all possible cross combinations of inbreds and testers which is equal to $593 \times 496 = 294, 128$. We randomly selected 100 inbreds and 100 testers and visualized the average yield performance of their cross combinations using heatmap. As shown in Fig 4, we can find the best cross combination for a given inbred or tester using the heatmap.

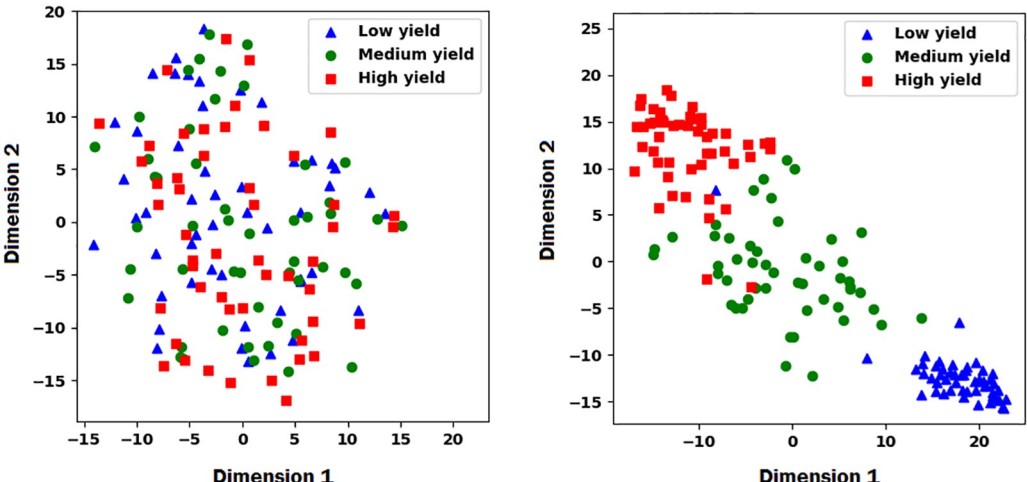

**Fig 3. The left and right plots show the t-SNE embedded plot of testers latent factors before and after training, respectively.** The plots illustrate how the proposed model was able to differentiate the high, medium, and low yield testers based on their corresponding latent factors in the embedding layer.

## Discussion

In this paper, we presented a collaborative filtering approach for predicting the yield performance of cross combinations of inbreds and testers that have not been planted based on the historical yield data collected from crossing other inbreds and testers together. The proposed model used an ensemble of the matrix factorization and the neural network to make predictions based on the inbreds and testers historical performances, their genetic grouping information, and planting locations. The proposed model was compared to other models such as DeepFM, LASSO, and random forest. The proposed model had a significantly better performance compared to models without any embedding layers such as LASSO and random forest because the embedding layers can help handle sparse one-hot input data. Compared to the DeepFM model, proposed model performed better for the following reasons: (1) the proposed model uses separate embeddings for the GMF and the neural network parts which would not limit the learning capacity of the model, and (2) the proposed model focuses more on the interactions between testers and inbreds and considers the location ID and the genetic grouping ID as auxiliary information. Similar collaborative filtering approaches have also been proposed in the literature. Cheng et al. [33] proposed a model which jointly trains wide linear models and deep neural networks to combine the benefits of learing low-order and high-order interactions for mobile app recommender systems. Covington et al. [34] proposed a deep learning based collaborative filtering model for YouTube Recommendations. Liu et al. [35] designed a deep hybrid recommender system framework based on auto-encoders, GMF and neural networks. Cui et al. [36] proposed a collaborative filtering approach for personalized point of interest recommendation. Their proposed approach addresses the problem of data sparseness by combining different methods such as preference mining, bi-relational hypergraph representation, and matrix factorization.

The computational results suggested that the proposed model was able to collaboratively learn the both low-order and high-order interactions between inbreds and testers and make reasonably accurate predictions. The proposed model can estimate the yield performance of any combination of inbreds and testers before actual crossings, which would help plant

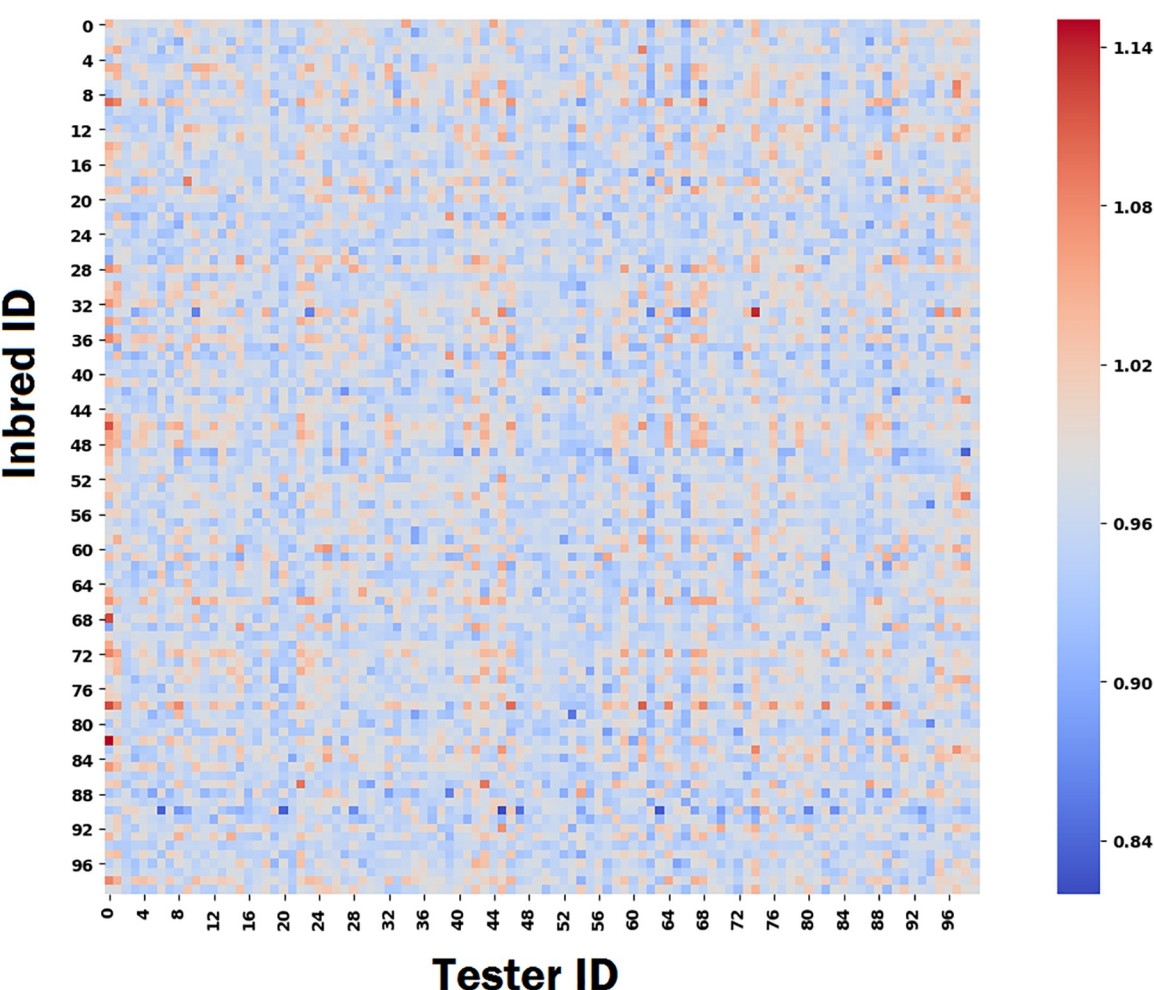

**Fig 4. The heatmap of the average yield performance of all cross combinations of 100 inbreds and 100 testers.**

breeders focus on the best possible combinations. This method could be extended to include other important variables such as weather components and soil conditions to improve the prediction performance.

## Acknowledgments

We thank Syngenta and the Analytics Society of INFORMS for organizing the Syngenta Crop Challenge and providing the valuable datasets. This manuscript has been released as a preprint at arXiv. The source code of the proposed method is available on GitHub [37].

## Author Contributions

**Conceptualization:** Saeed Khaki.

**Formal analysis:** Saeed Khaki, Zahra Khalilzadeh.

**Funding acquisition:** Lizhi Wang.

**Methodology:** Saeed Khaki, Zahra Khalilzadeh, Lizhi Wang.

**Software:** Saeed Khaki.

**Supervision:** Saeed Khaki, Lizhi Wang.

**Validation:** Saeed Khaki.

**Visualization:** Saeed Khaki, Zahra Khalilzadeh.

**Writing – original draft:** Saeed Khaki, Zahra Khalilzadeh, Lizhi Wang.

**Writing – review & editing:** Saeed Khaki, Lizhi Wang.

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
