## [Decision Letter · Decision Letter 0]

17 Mar 2020

PONE-D-20-03393

Predicting Yield Performance of Parents in Plant Breeding: A Neural Collaborative Filtering Approach

PLOS ONE

Dear Mr Khaki,

Thank you for submitting your manuscript to PLOS ONE. After careful consideration, we feel that it has merit but does not fully meet PLOS ONE’s publication criteria as it currently stands. Therefore, we invite you to submit a revised version of the manuscript that addresses the points raised during the review process.

We would appreciate receiving your revised manuscript by May 01 2020 11:59PM. To enhance the reproducibility of your results, we recommend that if applicable you deposit your laboratory protocols in protocols.io, where a protocol can be assigned its own identifier (DOI) such that it can be cited independently in the future. For instructions see: http://journals.plos.org/plosone/s/submission-guidelines#loc-laboratory-protocols

We look forward to receiving your revised manuscript.

Kind regards,

Le Hoang Son, Ph.D

Academic Editor

PLOS ONE

**Comments to the Author**

1. Is the manuscript technically sound, and do the data support the conclusions?

Reviewer #1: Partly

Reviewer #2: Yes

2. Has the statistical analysis been performed appropriately and rigorously? 

Reviewer #1: Yes

Reviewer #2: Yes

3. Have the authors made all data underlying the findings in their manuscript fully available?

Reviewer #1: Yes

Reviewer #2: Yes

4. Is the manuscript presented in an intelligible fashion and written in standard English?

Reviewer #1: Yes

Reviewer #2: Yes

5. Review Comments to the Author

**Reviewer #1**:

To predict the yield performance in corn hybrids, the authors propose an effective method, which combine neural network and collaborative filtering method. The method can estimate the yield performance of any combination of inbreds and testers before actual crossings, which would help plant breeders focus on the best possible combinations. The experiment results shows the excellent performance of the method.

Unfortunately, the key idea, combining the GMF and neural networks, has been proposed by Xiangnan He et al. in the paper “Neural Collaborative Filtering” 2017. In addition, the proposed framework in this paper is very similar to the He’s framework, it seems to lack of novelty. Besides this, apart from the ablation experiment methods, only two methods are selected as baseline methods, it will be better to find more effective methods to evaluate the performance.

Due to the above reasons, I do not think this work is excellent enough to be accepted.

Strong points:

In this paper, the authors propose a collaborative filtering approach for predicting the yield performance of cross combinations of inbreds and testers. The proposed model can estimate the yield performance of any combination of inbreds and testers before actual crossings, which would help plant breeders focus on the best possible combinations.

The computational results suggest that the proposed model was able to collaboratively learn the both low-order and high-order interactions between inbreds and testers and make reasonably accurate predictions.

This method could be extended to include other important variables such as weather components and soil conditions to improve the prediction performance.

Weak points:

The key idea, combining the GMF and neural networks, has been proposed by Xiangnan He et al. in the paper “Neural Collaborative Filtering” 2017. In addition, the proposed framework in this paper is very similar to the He’s framework. It seems that this work just apply He’s framework to predict the yield performance. It is lack of novelty.

The baseline methods are not appropriate. Firstly, apart from the ablation experiment methods, only two methods are selected as baseline methods. Secondly, these two methods are very fundamental and I do not think they are state-of-art methods. It will be better to find more effective methods to evaluate the performance.

Some important related work are missing in discussion, such as "Augmented Collaborative Filtering for Sparseness Reduction in Personalized POI Recommendation"..

The mathematical expression of loss function are not shown in the paper. I think this is a significant failure. The authors say they use Huber loss as loss function. It is well-known that the value of δ in Huber loss is an essential value however, I do not find it in the paper.

Typos and minors:

Figures and figure captions are not put together. Figures are in page 15, 16, 17, 18 while their captions are in other pages.

**Reviewer #2**:

The selection of an inbred and tester ID fits well in a collaborative filtering framework, and on the whole the paper is well written. However I have a few concerns with the formulation and the proposed solution.

1) While the question of breed selection is important, I am wondering if there is more to the problem that is missed in this formulation. Specifically, should we zoom into the genetic ID or location factor? It appears that a more nuanced formulation may be possible depending on the specifics of these factors. The dataset selection may also play a part in this choice, a discussion on whether additional fine-grained data (such as weather factors of a specific location or similar details of the genetic factors) might significantly improve our ability to make predictions would be useful. That is, does the proposed model manage to do fairly well even with just a Genetic ID and a Location ID instead of these specific details? It appears from a very brief overview, that Hammer, Graeme, et al. "Models for navigating biological complexity in breeding improved crop plants." Trends in plant science 11.12 (2006): 587-593. supports the idea of a course grained model doing very well, but a slightly nuanced argument might be useful to convince participants to adopt such a strategy.

2) On the solution side, the proposed framework is convincingly tested and the results are reproducible. I wonder if an additional dataset could be included to better validate the model, potentially with a slightly different combination of additional factors, to show that the framework generalizes to other data features. This also ties in to the previous suggestion regarding fine-grained descriptive factors. Maybe an ablation study where the specific details are anonymized to just an ID, and the overall model still performs in a similar manner would prove very convincing.

Despite having expressed these concerns, on the whole the main strengths of the paper are the simplicity and the clarity of the proposed architecture. Thus, I recommend acceptance with minor revisions to address the above two points.

---

## [Author Response · Author response to Decision Letter 0]

3 Apr 2020

Dear Editor,

We are grateful for your time and comments on our manuscript. We revised the manuscript to address the points raised during the review process. To enhance the reproducibility of our results, we uploaded the codes of our proposed method on GitHub and cited the repository in the acknowledgement section of the revised manuscript. We also used PLOS ONE journal latex template for the revised manuscript to meet PLOS ONE's style requirements.

We would like to also let you know that our work is selected as one of the 5 finalists for the 2020 Syngenta Crop Challenge by an independent judging committee (https://www.ideaconnection.com/syngenta-crop-challenge/judges.php) based on evaluation criteria such as:

1) Accuracy of the predicted values in the completely independent test set 

2) The quality and clarity of the submitted paper. 

The finalists will present their work on April 27, 2020 for judging committee, then judging committee will determine and announce the 2020 Syngenta Crop Challenge winners. 

Dear Reviewer 1,

To predict the yield performance in corn hybrids, the authors propose an effective method, which combine neural network and collaborative filtering method. The method can estimate the yield performance of any combination of inbreds and testers before actual crossings, which would help plant breeders focus on the best possible combinations. The experiment results shows the excellent performance of the method.

Thank you for your thoughtful and thorough review of our manuscript.

Unfortunately, the key idea, combining the GMF and neural networks, has been proposed by Xiangnan He et al. in the paper “Neural Collaborative Filtering” 2017. In addition, the proposed framework in this paper is very similar to the He’s framework, it seems to lack of novelty.

Thank you for your valuable comment. For the model selection part of this research, we tried many different models and neural network architectures. For example, we implemented models without embedding layers and also models with different types of embedding layers such as shared embedding. We tried models which are well-known to capture low-order interactions such as matrix factorization and also complex nonlinear models which are good at capturing high-order interactions such as deep neural networks. Finally, inspired by the papers “Neural collaborative filtering” and “Deepfm: a factorization-machine based neural network for CTR prediction” which are already cited in our paper, we implemented the proposed model in our paper which is a hybrid model to capture both low and high order interactions of testers and inbreds to improve the prediction accuracy. Both of these papers have an application in computer science domain such as movie and click-through rate recommendations. They have never been used in the agriculture domain. However, our proposed model has the following main differences with the paper “Neural collaborative filtering” proposed by Xiangnan He et al. 

1) “Neural collaborative filtering” paper deals with the implicit feedback recommendation, while our method is for the explicit feedback recommendation. 

2) “Neural collaborative filtering” paper does not include any axillary information (features related to movies and users such as age, sex, movie genre, etc.) and just focuses on the interactions between users and items. But, our proposed model includes auxiliary information such as genetic information and environments.

3) “Neural collaborative filtering” paper uses binary cross-entropy loss (log loss), while our model uses Huber loss which is a combination of L1 and L2 losses.

4) “Neural collaborative filtering” paper does not use dropout for regularization, our model uses dropout for avoiding overfitting and it improves the performance of the model.

Above-mentioned differences were added to the revised paper (highlighted sentences on page 3). 

We would like to point out that the key idea of combining the matrix factorization and neural networks with embedding have been used in many papers such as:

1) Deepfm: a factorization-machine based neural network for CTR prediction 

2) Neural collaborative filtering

3) Wide & Deep Learning for Recommender Systems

4) A Novel Deep Hybrid Recommender System Based on Auto-encoder with Neural Collaborative Filtering

All these papers used the idea for a specific application and all above-mentioned papers may have overlapping parts. 

We would also like to add that we do not claim the novelty completely in the method itself, but in the application in agriculture and plant breeding. To the best of our knowledge this is the first study to use the neural network based collaborative filtering method in plant breeding for cross yield prediction. We tested the approach on a very large amount of real data (199,476 observations) which suggested that this method could potentially help plant breeders to make better crossing decisions.

Besides this, apart from the ablation experiment methods, only two methods are selected as baseline methods, it will be better to find more effective methods to evaluate the performance.

Point well taken. We added two more baseline models to have better a comparison among the models. The new models are as follows:

1) DeepFM from the paper “Deepfm: a factorization-machine based neural network for CTR prediction” which is a popular model in collaborative filtering.

2) Factorization Machine method proposed by Rendle in 2010.

These two models are added in the section 4 of the revised paper on page 5. We updated the tables 1 and 2 in the results section (page 6, changes are highlighted) and we also explained the new results on page 7 (highlighted sentences).

Strong points:

In this paper, the authors propose a collaborative filtering approach for predicting the yield performance of cross combinations of inbreds and testers. The proposed model can estimate the yield performance of any combination of inbreds and testers before actual crossings, which would help plant breeders focus on the best possible combinations.

The computational results suggest that the proposed model was able to collaboratively learn the both low-order and high-order interactions between inbreds and testers and make reasonably accurate predictions.This method could be extended to include other important variables such as weather components and soil conditions to improve the prediction performance.

Thank you for acknowledging the strengths of our manuscript.

Weak points:

The key idea, combining the GMF and neural networks, has been proposed by Xiangnan He et al. in the paper “Neural Collaborative Filtering” 2017. In addition, the proposed framework in this paper is very similar to the He’s framework. It seems that this work just apply He’s framework to predict the yield performance. It is lack of novelty.

Thank you for your valuable comment. We already addressed this comment in your first comment.

The baseline methods are not appropriate. Firstly, apart from the ablation experiment methods, only two methods are selected as baseline methods. Secondly, these two methods are very fundamental and I do not think they are state-of-art methods. It will be better to find more effective methods to evaluate the performance.

Point well taken. We added two more baseline models to have better a comparison among the models. The new models are as follows:

3) DeepFM from the paper “Deepfm: a factorization-machine based neural network for CTR prediction” which is a popular model in collaborative filtering.

4) Factorization Machine method proposed by Rendle in 2010.

These two models are added in the Design of experiments section of the revised paper on page 6. We updated the tables 1 and 2 in the results section (page 7, changes are highlighted) and we also explained the new results on pages 7-8 (highlighted sentences).

Some important related work are missing in discussion, such as "Augmented Collaborative Filtering for Sparseness Reduction in Personalized POI Recommendation".

Thank you for your valuable suggestion. We added the following important related works in the discussion section of the revised paper (on page 10, highlighted sentences) :

1) Augmented Collaborative Filtering for Sparseness Reduction in Personalized POI Recommendation

2) Wide & Deep Learning for Recommender Systems

3) Deep Neural Networks for YouTube Recommendations

4) A Novel Deep Hybrid Recommender System Based on Auto-encoder with Neural Collaborative Filtering

The mathematical expression of loss function are not shown in the paper. I think this is a significant failure. The authors say they use Huber loss as loss function. It is well-known that the value of δ in Huber loss is an essential value however, I do not find it in the paper.

Point well taken. We added the mathematical expression of loss function in the revised paper (on page 5, equation 6). We also added the value of δ in Huber loss which is 0.10 in the revised paper (page 5).

Typos and minors:

Figures and figure captions are not put together. Figures are in page 15, 16, 17, 18 while their captions are in other pages.

Thanks for pointing out this comment. The PLOS One Journal requires figures to be placed at the end of the manuscript for review process that is why the captions and figures are not together.

Thank you again for your valuable comments and feedbacks which improved the quality of our paper.

In the end, we would like to mention that our work is selected as one of the 5 finalists for the 2020 Syngenta Crop Challenge by an independent judging committee (https://www.ideaconnection.com/syngenta-crop-challenge/judges.php) based on evaluation criteria such as:

1) Accuracy of the predicted values in the completely independent test set 

2) The quality and clarity of the submitted paper. 

The finalists will present their work on April 27, 2020 for judging committee, then judging committee will determine and announce the 2020 Syngenta Crop Challenge winners.

Dear Reviewer 2,

The selection of an inbred and tester ID fits well in a collaborative filtering framework, and on the whole the paper is well written. However I have a few concerns with the formulation and the proposed solution.

Thank you for your thoughtful and thorough review of our manuscript.

1) While the question of breed selection is important, I am wondering if there is more to the problem that is missed in this formulation. Specifically, should we zoom into the genetic ID or location factor? It appears that a more nuanced formulation may be possible depending on the specifics of these factors. The dataset selection may also play a part in this choice, a discussion on whether additional fine-grained data (such as weather factors of a specific location or similar details of the genetic factors) might significantly improve our ability to make predictions would be useful. That is, does the proposed model manage to do fairly well even with just a Genetic ID and a Location ID instead of these specific details? It appears from a very brief overview, that Hammer, Graeme, et al. "Models for navigating biological complexity in breeding improved crop plants." Trends in plant science 11.12 (2006): 587-593. supports the idea of a course grained model doing very well, but a slightly nuanced argument might be useful to convince participants to adopt such a strategy.

Thank you for your very insightful comment. Inspired by the paper proposed by Hammer, Graeme, et al. which suggests using coarse-grained models for phenotypic prediction in plant breeding, we excluded the fine details such as inbred ID and tester ID from the proposed model. We just used genetic grouping ID and location ID for the yield prediction which are high-level information. We added a section on page 8 in the revised paper (highlighted sentences, lines 264-277) to present the results of this new analysis.

The results indicated most of the variation in the crop yield is explained by the genetic grouping and the environmental factors. The results also suggested that a high level model based on only genetic grouping ID and location ID might be potentially useful for the yield prediction.

2) On the solution side, the proposed framework is convincingly tested and the results are reproducible. I wonder if an additional dataset could be included to better validate the model, potentially with a slightly different combination of additional factors, to show that the framework generalizes to other data features. This also ties in to the previous suggestion regarding fine-grained descriptive factors. Maybe an ablation study where the specific details are anonymized to just an ID, and the overall model still performs in a similar manner would prove very convincing.

Thank you for your valuable comment. These are valid points. We believe that adding additional data such as environmental factors and soil components would be very interesting thing to explore because we can have a performance comparison between coarse-grained and fine-grained models. But, unfortunately, the location ID were internally coded by Syngenta as proprietary information and they did not provide the latitude and longitude of these locations. As such, we cannot use other sources data for weather and soil since they require the coordinates of locations. 

Despite having expressed these concerns, on the whole the main strengths of the paper are the simplicity and the clarity of the proposed architecture. Thus, I recommend acceptance with minor revisions to address the above two points.

Thank you again for your valuable comments and feedbacks which improved the quality of our paper.

In the end, we would like to mention that our work is selected as one of the 5 finalists for the 2020 Syngenta Crop Challenge by an independent judging committee (https://www.ideaconnection.com/syngenta-crop-challenge/judges.php) based on evaluation criteria such as:

1) Accuracy of the predicted values in the completely independent test set 

2) The quality and clarity of the submitted paper. 

The finalists will present their work on April 27, 2020 for judging committee, then judging committee will determine and announce the 2020 Syngenta Crop Challenge winners.

---

## [Decision Letter · Decision Letter 1]

5 May 2020

Predicting Yield Performance of Parents in Plant Breeding: A Neural Collaborative Filtering Approach

PONE-D-20-03393R1

Dear Dr. Khaki,

We are pleased to inform you that your manuscript has been judged scientifically suitable for publication and will be formally accepted for publication once it complies with all outstanding technical requirements.

With kind regards,

Le Hoang Son, Ph.D

Academic Editor

PLOS ONE

**Comments to the Author**

1. If the authors have adequately addressed your comments raised in a previous round of review and you feel that this manuscript is now acceptable for publication, you may indicate that here to bypass the “Comments to the Author” section, enter your conflict of interest statement in the “Confidential to Editor” section, and submit your "Accept" recommendation.

Reviewer #1: All comments have been addressed

Reviewer #2: All comments have been addressed

2. Is the manuscript technically sound, and do the data support the conclusions?

Reviewer #1: Yes

Reviewer #2: Yes

3. Has the statistical analysis been performed appropriately and rigorously? 

Reviewer #1: Yes

Reviewer #2: Yes

4. Have the authors made all data underlying the findings in their manuscript fully available?

Reviewer #1: Yes

Reviewer #2: Yes

5. Is the manuscript presented in an intelligible fashion and written in standard English?

Reviewer #1: (No Response)

Reviewer #2: Yes

6. Review Comments to the Author

**EDITOR:**

Please upload (blinded) samples of experimental data while submitting the final manuscript for the sake of replication of your work. Besides, please ask a native English speaker to proofread the manuscript once again before re-submission. 

**Reviewer #1**: Authors have well-addressed all my concerns. I hence suggest to accept as is.

**Reviewer #2**:

I am recommending acceptance of this paper after the revision. While I understand that the original dataset cannot be directly shared owing to their agreement, an anonymized sample (added to the github repository) might be useful to other researchers who wish to test their models, at a preliminary stage.

---

## [Editor Report · Acceptance letter]

7 May 2020

PONE-D-20-03393R1 

Predicting Yield Performance of Parents in Plant Breeding: A Neural Collaborative Filtering Approach 

Dear Dr. Khaki:

I am pleased to inform you that your manuscript has been deemed suitable for publication in PLOS ONE. Congratulations! Your manuscript is now with our production department. 

With kind regards,

on behalf of

Prof. Le Hoang Son 

Academic Editor

PLOS ONE